# Differences in the Sensitivity of the Baroreflex of Heart Rate Regulation to Local Geomagnetic Field Variations in Normotensive and Hypertensive Humans

**DOI:** 10.3390/life12071102

**Published:** 2022-07-21

**Authors:** Liliya Poskotinova, Elena Krivonogova, Denis Demin, Tatyana Zenchenko

**Affiliations:** 1N. Laverov Federal Center for Integrated Arctic Research of the Ural Branch of the Russian Academy of Sciences, 163069 Arkhangelsk, Russia; elena200280@mail.ru (E.K.); denisdemin@mail.ru (D.D.); 2Institute of Theoretical and Experimental Biophysics, Russian Academy of Sciences, 142290 Pushchino, Russia; zench@mail.ru; 3Space Research Institute, Russian Academy of Sciences, 117997 Moscow, Russia

**Keywords:** heart rate variability, geomagnetic field, baroreflex

## Abstract

Synchronization between heart rate variability (HRV) in the low-frequency (LF) range (0.04–0.15 Hz) and 1-min variations in the components (X, Y, Z)and the total vector (F) of geomagnetic induction (nT) was studied in normotensive (blood pressure up to 140/90 mmHg) and hypertensive (blood pressure above 140/90 mmHg) individuals living in the Arkhangelsk region (60°51′52″ N 39°31′05″ E).The duration of registration of HRV for each person is 30 min in a sitting position. The most pronounced synchronization of the LF parameter, which reflects baroreflex activity, with variations in the GMF was found in normotensive individuals. The absence of a significant synchronization of the LF parameter with variations in the GMF components indicates a decrease in the sensitivity of the baroreflex mechanism and a risk of dysregulation of vascular tone, especially in people with arterial hypertension, under conditions of instability of the geomagnetic field.

## 1. Introduction

At high latitudes on Earth, including the Arctic zone of the Russian Federation, a special electromagnetic background is formed, where the wave structure of magnetic variations causes the formation of polar substorms. [1]. The diversity of electromagnetic phenomena in all layers of near-Earth space in the Arctic also determines the presence of a wide range of individual adaptive responses of human physiological systems [2]. The direction of the magnetic field vector of the heart depends on the state of health of a living organism. According to the surface mapping of cardiac bioelectric potentials, changes occur in the dynamics of the displacement of areas of positive and negative cardioelectric potentials during the period of atrial contraction, indicating the heterogeneity of the propagation of the excitation wave from the sinoatrial node to the lower parts of the heart during the development of arterial hypertension [3]. According to the assessment of the autonomic regulation of cardiac activity in an external magnetic field, it was shown that, for the human body, the most intense stimuli have a low frequency and intensity. With an increase in stimulation strength to average values, the tension of cardiac activity significantly decreases and remains at the same level, despite a further increase in frequency or intensity to high values [4]. That is, weak stimuli of the geomagnetic field can produce a significant biological response. At the same time, in healthy individuals, the magnetic field can act as both a synchronizer and a desynchronizer of the autonomic regulation of the heart rhythm. However, the search continues for methodological approaches to determining a “physiological measure” of the reactivity of cardiac activity, not for the entire population of people, but for each person individually. The idea of desynchronization of rhythmic processes in the geomagnetic environment and the human cardioelectric field as the main mechanism of cardiovascular pathology has also been developed [5]. That is, pronounced variations in the values of both the vertical and horizontal components of the geomagnetic field (GMF) can cause disturbances in the biorhythm of the physiological parameters of a person with high weather sensitivity. When considering the relationship between geophysical and biological processes, the modern literature pays more attention to the resonance mechanism. This mechanism is successfully used in medicine during transcranial magnetic stimulation, with the selection of a certain frequency of magnetic stimulation causing a synchronized response of neuronal activity—for example, in the motor cortex of the brain [6]. The studies by Borodin et al. have shown that heart rate variability indicators respond to variations in the eastern component (Y) of the GMF, ambient temperature, relative humidity, atmospheric pressure, and amplitude of the first Schumann resonance mode [7]. The frequency of most biotropic fluctuations in the activity of parameters of the cardiovascular system ranges from 0.001 to 10 Hz. Violations of the biorhythmogenesis of human regulatory systems in this range, caused by external factors, among other things, determine the risk of cardiac pathology in humans.

The autonomic regulation of the cardiovascular system may be impaired, resulting in an increased risk of arterial hypertension in humans. One of the mechanisms of arterial hypertension is a decrease in the sensitivity of baroreceptors, leading to an increase in blood pressure in the elderly, especially women [8,9]. Irritation of the baroreceptors of large vessels with an increase in stroke volume and vascular distension causes a decrease in heart rate, which reduces the risk of damage and rupture of the vessel wall layers during stress. Reduced spontaneous baroreflex activity causes baroreflex dysfunction [10] and resistance to antihypertensive therapy. In contrast, an increase in baroreflex reactivity is associated with the effectiveness of antihypertensive therapy and maintaining stable blood pressure at target levels [11].

Gmitrov showed that when a person is exposed to a static artificial magnetic field, the baroreflex reaction of the vessels increases, as does the intensity of microcirculation. When the natural geomagnetic field is disturbed, the baroreflex reaction, on the contrary, weakens, which is reflected in the weakening of the blood circulation [12]. It has been argued that natural variations in the geomagnetic field modulate the central nervous system’s mechanisms of regulation of the realization of the baroreflex, reducing the direct stimulating effect of artificial magnetic fields on the carotid sinus receptors and the vasodilatation reaction mediated by the baroreflex [12].

Violations of the biorhythmic genesis of the regulatory systems of the human body, caused by external geomagnetic factors, among other things, determine the risk of cardiac pathology, especially in the elderly. It has been shown that the frequency of most biotropic fluctuations in the activity of parameters of the cardiovascular system ranges from 0.001 to 10 Hz. A low frequency of heart rate variability (HRV) is associated primarily with the baroreflex regulation of heart rate [13,14]. With the continuous monitoring of HRV indicators for 7 days in healthy residents of the Arctic (Norway), it was shown that during periods of geomagnetic disturbances, the average daily heart rate significantly increases (by 7–8%), and the total HRV power decreases (by 18–19%), mainly in the range of low and very low HRV frequencies. The low-frequency (LF) component of the HRV spectrum decreases during periods of geomagnetic disturbances, which may cause the dysregulation of blood pressure during these periods [15]. It is assumed that the nature of the synchronization of changes in the low-frequency HRV and variations in the components of the local geomagnetic field (X, Y, Z) and the full vector of magnetic induction (F) will differ between normal and hypertensive individuals. Therefore, the purpose of this study is to determine the nature of the synchronization between the low-frequency indicator of HRV and variations in the components of the local GMF in people with normal and high blood pressure (BP).

## 2. Materials and Methods

A survey of a group of local residents (n = 41; 35 women and 6 men) was carried out in the Arkhangelsk region of the Russian Federation (60°51′52″ N 39°31′05″ E) from 13:30 on 18 March 2019, to 19:00 on 21 March 2019, Moscow time (UTC+3). The three-hour values of the planetary Kp index ranged from 0 to 3. An observatory of the Geophysical Center of the Russian Academy of Sciences (Klimovskaya, or KLI) was installed in the study area for monitoring the geomagnetic activity. The criterion for inclusion in the sample was voluntary participation in the study based on informed consent to participate in the study. The criteria for exclusion from the sample were cardiac arrhythmias, acute myocardial infarction, cerebrovascular accidents, epilepsy, neurodegenerative diseases, and signs of acute inflammatory processes (including influenza, viral infection, etc.). In total, 15 individuals (13 women, 2 men) had a normal blood pressure (no more than 140/90 mmHg),called Group I (age 46.9 ± 10.5 years), and 26 people (22 women, 4 men) had a blood pressure above 140/90 mmHg, called Group II (age 51.3 ± 11.8 years). Six out of 26 people with signs of an elevated blood pressure were not compliant to antihypertensive treatment, and 20 people were taking selective beta-blockers and angiotensin-converting enzyme inhibitors. Systolic and diastolic blood pressure was recorded using a digital blood pressure monitor (A&D, Tokyo, Japan). The age values in the groups are statistically identical (*p* > 0.05).

A cardiointervalogram using the Varicard 2.6 equipment (LLC Ramena, Ryazan, Russia) was recorded for each participant for 30 min in a sitting position. According to the recommendations of the North American Society for Electrical Stimulation and Electrophysiology, HRV data of at least 2 min of recording were used to assess the LF parameter. In order to obtain the most representative variational range of LF values in the subsequent 30-min series of cardiointervalograms used by the Varicard equipment, the series were divided into 2-min segments with a 1-min overlap (see Figure 1).

This made it possible to obtain 29 LF values (0.04–0.15 Hz) [4,13,14] in absolute values (ms^2^) and as a percentage (%) of the total HRV for subsequent correlation analysis. The first recording period (0–2 min) was not taken into account as a primary non-stationary process. Subsequently, the LF value from 1 to 3 min of HRV recording was regarded as the first point, the value from 2 to 4 min was regarded as the second point, and so on, until the 28th point (from 28 to 30 min); there are 28 points in the variation series for each person. The 1-min variations in the values of the geomagnetic field (GMF) induction—the X, Y, Z components of the GMF vector (nT) and the total vector (F) of geomagnetic induction (nT)—were obtained from the website of the Geophysical Center of the Russian Academy of Sciences (http://geomag.gcras.ru/obs-KLI.html; accessed on 1 April 2022) via the Klimovskaya station (KLI). The time of the beginning of HRV registration for each person and several values of GMF variations in the X, Y, Z components of the total vector of magnetic induction F along the Greenwich meridian were used.

Correlation coefficients were calculated between all possible pairwise combinations of biological (LF, ms^2^, and LF%) and geomagnetic (X, Y, Z, and F) series of values for each person separately. Mathematical operations of frequency filtering were applied to the analyzed time series to exclude certain periods using the MATLAB program (version R2010a). Before performing a correlation analysis due to removing low-frequency trends from the signal, the time series were passed through a band-pass filter with a Blackman–Harris window with an upper cutoff frequency *fr* = 0.9995 from the Nyquist frequency and a lower cutoff frequency *fl *= 0.70. A matrix of Spearman correlation coefficients, with time series of 1-min values of geomagnetic induction by the GMF components obtained from the Klimovskaya station (KLIX, KLIY, KLIZ, KLIF),was calculated with a zero time lag (synchronous values) between the physiological and geophysical time series (*p* < 0.05). Comparison of the percentages of individuals with different BP levels and significant correlation coefficients between GMF and HRV was performed using Fisher’s exact test (*p* < 0.05). Thus, the data analysis design included the following steps: (1) formation of 1-min time series of variations in the values of the geomagnetic field components and 2-min overlapping 2-min series of LF values (in ms^2^ and LF%), (2) filtering the time series to exclude low-frequency trends; (3) correlation analysis separately for each person of time series of geomagnetic field components and LF values (in ms^2^ and %); and (4) assessment of the proportion of persons with significant correlations in groups of normotensive and hypertensive people.

## 3. Results

It has been established that the synchronization of LF values with GMF variations is of an individual nature; that is, for a particular person, synchronization of LF with variations in magnetic induction can be either in one or several components of the GMF. For further analysis, cases were considered when, out of all the values of the GMF components, there was at least one statistically significant correlation (*p* < 0.05) with the LF values. Figure 2 shows an example of the synchronization of LF% values and magnetic induction values according to the Y component of the GMF in participant No.16 (a 45-year-old woman, BP 120/80 mmHg).

Figure 3 shows an example of the absence of significant synchronization of LF values and magnetic induction for any of the GMF components—in particular, for the Y component in participant No. 32 (woman, 60 years old, BP 166/90 mmHg). It is shown here that from 19 to 29 min of observation, fluctuations in baroreflex activity (LF) with fluctuations in GMF variations are not expressed. This indicates a low baroreflective sensitivity to GMF variations in this person. In general, almost all persons with normal BP (14 out of 15), with the exception of one person, had a significant correlation of LF values with variations in one of the GMF components. An additional analysis revealed that this person had an anamnesis of a drop in blood pressure to 90/60 mmHg, which can be regarded as a violation of vascular tone by the mechanism of adrenergic deficiency. Therefore, after the exclusion of this person, the group of people with normal blood pressure consisted of 14 persons.

According to Table 1, in all people with a normal blood pressure (Group I), the LF values, expressed either in percentage or absolute terms, are synchronized with the values of at least one of the GMF components, whereas in people with a high blood pressure (Group II), such synchronization occurs significantly less often—in a little more than half of the cases.

Thus, a more pronounced synchronization of the indicator reflecting baroreflex activity with variations in the local geomagnetic field was found in people with a normal blood pressure. 

## 4. Discussion

In persons with normal nervous regulation of vascular tone, fluctuations in the magnitude of the magnetic field induction cause amplitude fluctuations in heart rate, which ensures the preservation of the heart rate variability, reflecting the functional reserve of the autonomic regulation of cardiac activity. Violations of the nervous regulation of vascular tone cause heterogeneity and instability of the electromagnetic field of the heart, which leads to an increase in its vulnerability to external magnetic fields. According to the assessment of the autonomic regulation of cardiac activity in an external magnetic field, it was shown that, for the human body, the most intense stimuli have a low frequency and intensity. With an increase in the strength of stimulation, the tension of cardiac activity may remain at the same level, despite a further increase in the frequency or intensity of magnetic field exposure [4]. That is, weak magnetic stimuli, close to the strength of a natural magnetic field, can give a significant biological response.

Other authors have previously shown that the highest degree of synchronization of heart rate with variations in the geomagnetic field was observed on the geomagnetically most quiet day (Ap index = 1), whereas the lowest degree was observed on the day of geomagnetic disturbances (Ap index = 32) [16]. This confirms the importance of detecting the synchronization of heart rate variability and geomagnetic fluctuations on days with a calm geomagnetic situation. When assessing atmospheric changes (humidity, atmospheric pressure, wind speed) and the geomagnetic field (X component of the GMF) during the modulation of the cardiovascular system, we found that, in healthy people, changes in heart rate most often caused variations in the values of the X components of the GMF [16].

The method presented here made it possible to reveal differences in the synchronization of the HRV indicator, which reflects baroreflex activity, and GMF variations in humans even against a background of an undisturbed geomagnetic field. It is known that the rhythmicity of fluctuations in the activity of bioelectric tissues is associated with periods of fluctuations in the concentrations of calcium, magnesium, and chloride ions, which, in turn, are synchronized with periods of variations in the X and Y components of the geomagnetic field [17]. The opinion of other authors that the absence or loss of significant correlations between GMF variations and cardiac activity might pose a risk of cardiovascular pathology is confirmed [18]. 

One of the mechanisms of disturbances in the sensitivity of the baroreflex to external environmental influences may be a decrease in endothelial nitric oxide, which, in turn, is associated with the pathogenesis of arterial hypertension. At the same time, the importance of identifying these disorders at the preclinical stage is emphasized. Baroreflex-mediated increment in vessel sensitivity to nitric oxide is suggested to be a new mechanism in baroreflex physiology, with potential implementation in cardiovascular conditions in which nitric oxide deficit and autonomic dysfunction substantially increase the risk of morbidity and mortality [19].

The change in the synchronization of the low-frequency part of the HRV in people with arterial hypertension may be based on a violation of the mechanisms of cyclotron resonance, which ensures the natural magnetic sensitivity of the tissues of the human body. With variations in the external magnetic field, the movement of ions occurs along a cycloidal trajectory in the plane of the magnetic field perpendicular to the main direction and with a certain circular (cyclotron) frequency, the value of which depends on the charge, ion mass, and field strength. Ion cyclotron resonance can lead to molecular changes associated with enhanced proton transport [20,21]. Despite the fact that the geomagnetic field is weak for transferring an energy impulse to subcellular particles, receptors, or individual ions (in comparison with the energies of thermal fluctuations), the resonance-like response of a cell to variations in the geomagnetic field may nevertheless be one of the molecular mechanisms for maintaining baroreflex activity in people with normal regulation of vascular tone. The hypothesis, that with aging there is a decrease in the resonant response of the cell to variations in the local geomagnetic field, which generally causes a decrease in baroreflex activity, needs further experimental verification.

Thus, the presented method for determining the degree of synchronization between GMF variations and the low-frequency part of HRV on 30-min recordings of the cardiorhythmogram can be used to assess the adaptive capabilities of the human cardiovascular system to the instability of the geomagnetic field. Data on the sensitivity of baroreflex regulation of heart rate to variations in the local geomagnetic field in humans can be used for corrective procedures using artificial magnetic fields to optimize baroreflex activity [22], which is especially important for the elderly.

It is also possible to use this method to identify individuals who need to activate the baroreflex with slow breathing. In this case, there is an increase in the value of heart rate variability at a resonant frequency of 0.1 Hz (including in low frequency of HRV). An increase in the low-frequency part of HRV is accompanied by improved vagally mediated heart rate variability and baroreflex sensitivity. These effects are achieved through temporal coherence of respiratory, blood pressure, and cardiac phases [23].

Currently, much attention is being paid to studying the multiscale structure of HRV series using nonlinear analysis methods. This may indeed provide qualitatively new information on the relationship between the mechanisms of heart rhythm regulation [24] and variations in the geomagnetic field vector components. In further studies, provided the series duration is increased (more than 30 min) and/or 1-s geophysical data are obtained, we plan to use nonlinear parameters of heart rhythm variability analysis. 

## 5. Conclusions

A new model for detecting the level of personal baroreflex sensitivity (according to the low-frequency part of the heart rhythm variability) to variations in the local geomagnetic field in humans is presented. For the first time, it is shown that synchronization of the values of the low-frequency portion of HRV (LF) and variations in the values of the local GMF components in hypertensive people is significantly reduced during 1-min variations in the values of the GMF components. The presence of a significant correlation of LF or LF% with magnetic induction fluctuations in one of the components of the local GMF (X, Y, Z) or the full vector of magnetic induction F indicates sufficient synchronization of the baroreflex activity with variations in the GMF and a relatively low risk of cardiovascular disorders associated with geomagnetic disturbances. The absence of the relationship between the above parameters indicates a weak synchronization of baroreflex activity with GMF variations and a risk of cardiovascular disorders during periods of geomagnetic disturbances, especially in hypertensive individuals.

The limitations of the study included the limitations in the duration of recording heart rate variability, the dimension of variation series of heart rate variability, and variations in the geomagnetic field components, which did not allow the full use of the multiscale entropy method for analyzing heart rate variability.

## Figures and Tables

**Figure 1 life-12-01102-f001:**
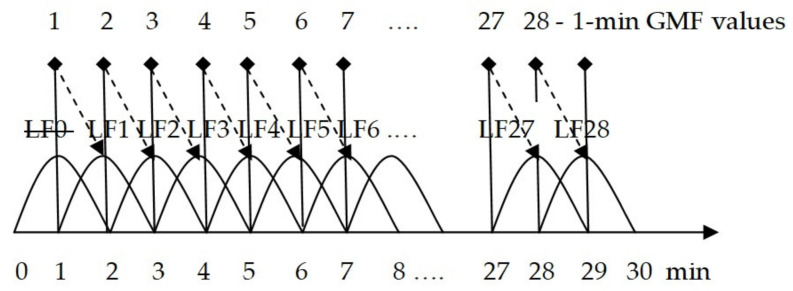
The ratio of the registration time of 2-min LF indicators and 1-min GMF values.The dotted line indicates the indicators for correlation analysis; the crossed-out values (LF0) were not used in the analysis.

**Figure 2 life-12-01102-f002:**
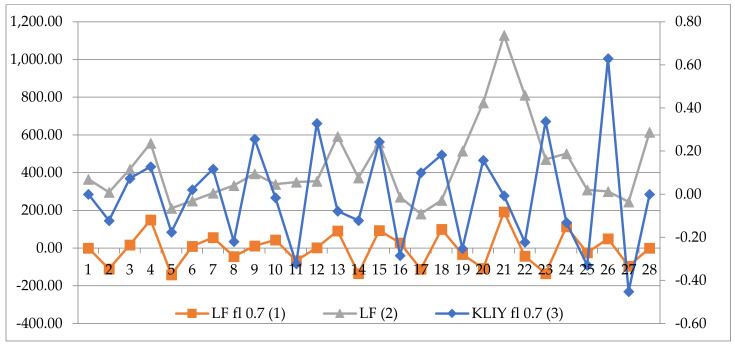
An example of a correlation (Spearman, *p* < 0.05) between LF values and magnetic induction in the Y component of the GMF in participant No. 16 (a 45-year-old woman, BP 120/80 mmHg); LF *fl* = 0.7 (1): data after filtering with *fl* = 0.7; LF (2): native LF data (in ms^2^); KLIY *fl* 0.7 (3): data of the Y component of the GMF after filtering with *fl* = 0.7.

**Figure 3 life-12-01102-f003:**
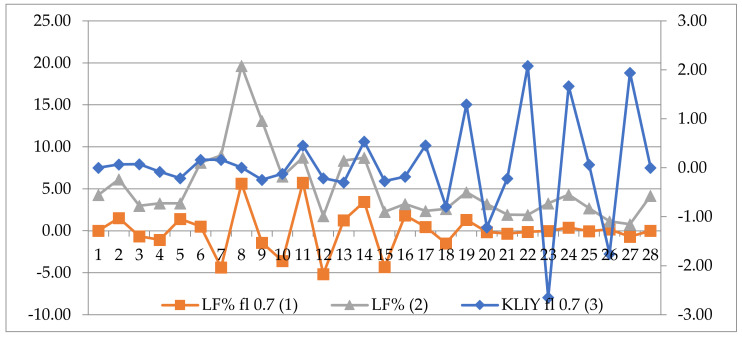
An example of the absence of a correlation (Spearman, *p* > 0.05) between the LF% values and magnetic induction in the Y component of the GMF in participant No. 32 (60-year-old woman, BP 161/94 mmHg); LF% *fl* 0.7(1): data after filtering with *fl *= 0.7; LF% (2): native LF data (in %); KLIY *fl* 0.7 (3): data of the Y component of the GMF after filtering with *fl *= 0.7.

**Table 1 life-12-01102-t001:** Frequency of occurrence (quantity/%) of significant correlations (Spearmen, *p* < 0.05) of LF values with the values of one of the GMF components (X, Y, Z, or F) among normotensive and hypertensive individuals after filtering the data series (*fl* = 0.7).

Variables	Group I (n = 14)	Group II (n = 26)	*p*
LF, mc2	12/85.7	17/65.4	0.170
LF, %	11/78.6	15/57.7	0.187
LF, mc2 or LF, %	**14/100**	**18/69.2**	**0.034**

## Data Availability

Not applicable.

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
