# Peer review of "Differences in the Sensitivity of the Baroreflex of Heart Rate Regulation to Local Geomagnetic Field Variations in Normotensive and Hypertensive Humans"

_life, 2022, doi:10.3390/life12071102_

Round 1

Reviewer 1 Report

Summary and general comments

 In this paper, the authors are proposing the most pronounced synchronization of the LF parameter which reflects baroreflex activity, with variations in the GMF found in normotensive individuals.

As we know, SDNN, RMSSD, and PNN50 are time-domain parameters of HRV. Nonlinear domain parameters of HRV (e.g., small scale multiscale entropy index (not ApEn)) were not included in the manuscript. The authors must also describe the methods in detail. Please focus on “new” in your study compared with others (model, design, or idea?).

Although I find the result a little interesting, I have significant reservations about some technical and scientific aspects of the current study:

Major issues

1. The title of the manuscript (e.g., general statement) needs to be modified to fit the “new” in your study compared with others (model, design, idea, application, findings, conclusions?).

2. In lines 124-153, the HRV statement should be more references included in the introduction section.

3. The authors must also describe the methods in detail. Please add a small scale multiscale entropy index or multiscale Poincaré index (refer to PMID: 30415711 DOI: 10.1016/j.cmpb.2018.10.001) to replace ApEn.

4. In table I (line:194,195) Group I(n=14), and Group II(n=26), where are different from a survey of a group of local residents (n=41) in line 99.

     Which one is correct?

5. The words in the Conclusions section need to be rewritten.

Minor issues

  I suggest that the authors carefully go through the paper again and correct all typos.

Author Response

 We are grateful to the referee for a detailed analysis of our scientific results.

We also had the article proofread in English (British) by staff of the professional firm (ProofreadingServices.com), of which we have written confirmation.

  1. The title of the manuscript (e.g., general statement) needs to be modified to fit the “new” in your study compared with others (model, design, idea, application, findings, conclusions?).

Response: We've updated the title: Differences in the sensitivity of the baroreflex of heart rate regulation to local geomagnetic field variations in normotensive and hypertensive humans

  1. In lines 124-153, the HRV statement should be more references included in the introduction section.

Response: We have added references in this part of text, included in the introduction section ([4, 13, 14]) - (highlighted in colour)

  1. The authors must also describe the methods in detail. Please add a small scale multiscale entropy index or multiscale Poincaré index (refer to PMID: 30415711 DOI: 10.1016/j.cmpb.2018.10.001) to replace ApEn.

Response:

We have added more detail to the study design (marked in colour). Thus, the data analysis design included the following steps: 1) formation of 1-minute time series of variations in the values of the geomagnetic field components and 2-minute overlapping 2-minute series of LF values (in ms2 and LF%), 2) filtering the time series to exclude low frequency trends; 3) correlation analysis separately for each person of time series of geomagnetic field components and LF values (in ms2 and %); 4) assessment of the proportion of persons with significant correlations in groups of normotensive and hypertensive people.

Nonlinear methods for analyzing time series (and, in particular, series of RR intervals) mentioned by the reviewer have been proposed and developed in the last 20 years. These methods have indeed shown high efficiency in the task of non-invasive diagnosis of diseases [1] and distinguishing between cohorts (for example, smokers and non-smokers [2], people with and without sleep apnea [3], people with diabetes and healthy people [4], etc. .). We also agree with the reviewer that the use of the MSE method significantly expands the possibilities for studying the complexity and nonlinear features of biological time series [5].

However, we refrained from using these methods in our study for the following reasons.

  • At first, in our study, the number of normotensive and hypertensive people was initially known before the start of statistical analysis. Also, we did not have the task of estimating the nonlinear complexity of a number of RR-intervals in a sample of individuals. We consider this to be a separate task for future research. At present, the task of our study was to investigate a new phenomenon of synchronization of the baroreflex mechanism, which we determined by the low-frequency part of the HRV, with variations in the local components of the GMF over a limited recording period (30 minutes). We tried to take into account the well-known conservatism of medical specialists in the approaches and methods of experiment and analysis used. We tried to find out whether there are differences in the characteristics of the response of the heart rate in groups of individuals with different level of blood pressure to variations in the components of the local geomagnetic field. Previously, differences were repeatedly observed in the amplitude of the response of blood pressure to the effects of magnetic storms, taking into account the planetary indices of geomagnetic activity - Ap or Dst index [6]. Also, our goal was to convey this information to physiologists who deal with the biological effects of space weather, and physicians who are involved in the prevention and treatment of meteosensitivity. Therefore, we purposefully chose the classic, recognized for more than 50 recent years, HRV indicators. In our case, the advantage of these indicators was, firstly, the presence of their generally accepted physiological interpretation, and secondly, their understandability and familiarity for practicing physicians, which would facilitate their perception of our scientific results. Of course, there are studies pointing to a possible physiological interpretation of the Poincare indices (SD1, SD2 and SD1/SD2) [7, 8]. However, the introduction of these indicators into the practice of a doctor is not yet so widespread, in contrast to the "classical" HRV indicators.
  • Secondly, we were limited in the choice of analysis methods by the duration and format of the experimental time series. Since the data on the components (X, Y, Z) of the local geomagnetic field vector (F) are freely available only with a sampling rate of 1 time per minute, we could not analyze biological data with a higher frequency in this task. With a row length of 30 minutes, the use of MSE would be incorrect. That is why we did not choose the task to investigate the complexity of the variational series of data on different time scales.
  • In general, we understand the importance of the reviewer's position that the study of the multiscale structure of the HRV series can indeed provide qualitatively new information about the relationship between the mechanisms of heart rate regulation [9], including with variations in the components of the geomagnetic field vector. In further studies, subject to increasing the duration of the series (more than 30 minutes) and/or obtaining 1-second geophysical data, we plan to apply the above methods of analysis. We have added this information as well as 1 reference in the reference list at the end of the Discussion section (highlighted in colour).
  • We have included information in the limitations of the study at the end of article. Limitations of the study included limitations in the duration of recording heart rate variability, the dimension of variation series of heart rate variability and variations in the geomagnetic field components, which did not allow the full use of the multiscale entropy method for analyzing heart rate variability (highlighted in colour).

  1. Frassineti L, Lanatà A, Olmi B, Manfredi C. Multiscale Entropy Analysis of Heart Rate Variability in Neonatal Patients with and without Seizures. Bioengineering (Basel). 2021 Sep 9;8(9):122. doi: 10.3390/bioengineering8090122. PMID: 34562944; PMCID: PMC8469929.
  2. Bagus Haryadi, Po-Hao Chang, Akrom Akrom, Arifan Q. Raharjo, Galih Prakoso Poincaré plots to analyze photoplethysmography signal between non-smokers and smokers. International Journal of Electrical and Computer Engineering (IJECE) Vol. 12, No. 2, April 2022, pp. 1565~1570 ISSN: 2088-8708, DOI: 10.11591/ijece.v12i2.pp1565-1570
  3. Wen-Yao Pan, Mao-Chang Su, Hsien-Tsai Wu, Meng-Chih Lin, I-Ting Tsai and Cheuk-Kwan Sun Multiscale Entropy Analysis of Heart Rate Variability for Assessing the Severity of Sleep Disordered Breathing. Entropy201517(1), 231-243; https://doi.org/10.3390/e17010231
  4. Haryadi B, Liou JJ, Wei HC, Xiao MX, Wu HT, Sun CK. Application of multiscale Poincaré short-time computation versus multiscale entropy in analyzing fingertip photoplethysmogram amplitudes to differentiate diabetic from non-diabetic subjects. Comput Methods Programs Biomed. 2018 Nov;166:115-121. doi: 10.1016/j.cmpb.2018.10.001. Epub 2018 Oct 2. PMID: 30415711.
  5. Humeau-Heurtier A. Multiscale Entropy Approaches and Their Applications. Entropy (Basel). 2020 Jun 10;22(6):644. doi: 10.3390/e22060644. PMID: 33286416; PMCID: PMC7517182.
  6. Zenchenko, T.A.; Breus, T.K. The Possible Effect of Space Weather Factors on Various Physiological Systems of the Human Organism. Atmosphere 2021, 12, 346. https://doi.org/10.3390/atmos12030346
  7. Peter Walter Kamen, Henry Krum, Andrew Maxwell Tonkin; Poincaré Plot of Heart Rate Variability Allows Quantitative Display of Parasympathetic Nervous Activity in Humans. Clin Sci (Lond)1 August 1996; 91 (2): 201–208. doi: https://doi.org/10.1042/cs0910201
  8. Brennan M, Palaniswami M, Kamen P. Poincaré plot interpretation using a physiological model of HRV based on a network of oscillators. Am J Physiol Heart Circ Physiol. 2002 Nov;283(5):H1873-86. doi: 10.1152/ajpheart.00405.2000. PMID: 12384465.
  9. Costa M., Goldberger A.L., Peng C.-K. Multiscale entropy analysis of biological signals.  Rev. E. 2005;71:021906. doi: 10.1103/PhysRevE.71.021906

  1. In table I (line:194,195) Group I(n=14), and Group II(n=26), where are different from a survey of a group of local residents (n=41) in line 99.

     Which one is correct?

Response: In the Results section, we initially indicated that 1 person was excluded from the analysis. In general, almost all persons with normal BP (14 out of 15), with the exception of 1 person, had a significant correlation of LF values ​​with variations in one of the GMF components. An additional analysis revealed that this person had an anamnesis of a drop in blood pressure to 90/60 mm Hg, which can be regarded as a violation of vascular tone by the mechanism of adrenergic deficiency. After the exclusion of this 1 person, the group of people with normal blood pressure was 14 people. Therefore, in the table we used n=14 and n=26.

  1. The words in the Conclusions section need to be rewritten.

Response: We have added information to the conclusion section, making the accent on new results (new model and new findings). A new model for detecting the level of personal baroreflex sensitivity (according to the low-frequency part of the heart rhythm variability) to variations in the local geomagnetic field in humans is presented.  For the first time, it is shown that synchronization of the values of the low-frequency portion of HRV (LF) and variations in the values of the local GMF components in hypertensive people is significantly reduced during 1-minute variations in the values of the GMF components. 

Reviewer 2 Report

the English language needs to be improved. the topic is very current and interesting with a wide implication in clinical practice.

Author Response

We are grateful to you for your work in reviewing our article. We have proof-read the English version of the text and made the appropriate additions.

We have improved the list of references, the discussion of results and the conclusion, and added a section on the limitations of the study. We have highlighted all additions in the new text.

Reviewer 3 Report

A nice manuscript. Can improved presentation for Table 1 by adding more columns. To note that p 0,03 should be p=0.03.

Author Response

We are grateful for your work in reviewing our article. According to your request, we added an additional column to the table and corrected the table data. We have also added information to the discussion section of the results and conclusion, which has been highlighted in colour in the new version of the article.

This manuscript is a resubmission of an earlier submission. The following is a list of the peer review reports and author responses from that submission.